# Soil Texture Explains Soil Sensitivity to C and N Losses from Whole-Tree Harvesting in the Boreal Forest

Rock Ouimet [1,*], Nathalie Korboulewsky [2] and Isabelle Bilger [2]

1   Direction de la Recherche Forestière, Ministère des Ressources Naturelles et des Forêts du Québec, Complexe Scientifique, 2700 Einstein, Quebec, QC G1P 3W8, Canada
2   INRAE—UR Écosystèmes Forestiers, Domaine des Barres, 45290 Nogent-sur-Vernisson, France
*   Correspondence: rock.ouimet@mffp.gouv.qc.ca; Tel.: +1-418-643-7994

**Abstract:** The use of forest biomass to produce energy is increasingly viewed as a means to reduce fossil fuel consumption and mitigate global warming. However, the impact of such practices on soils in the long term is not well known. We revisited forest plots that were subjected to either whole-tree (WTH, n = 86) or stem-only (SOH, n = 110) harvesting 30 years ago in the boreal forest in Quebec, Canada. The objective of the present study was to find soil properties that could explain the lower soil C and N stocks at the sites subjected to WTH compared to SOH after 30 years. Compared to SOH, lower soil C and N stocks attributable to WTH occurred when soil particle content <20 μm was below 30%. The theoretical separation of soil organic matter into two fractions according to soil particle content <20 μm—a recalcitrant and a labile fraction—could explain the observed pattern of soil C and N differences between WTH and SOH. Imperfect or poor soil drainage conditions were also associated with lower soil C and N in WTH compared to SOH. Limiting additional biomass harvesting from these sites would help to preserve soil C and N from potential losses.

**Keywords:** boreal forest soils; carbon; drainage; mineral-associated organic matter; nitrogen; soil fine particles; whole-tree harvesting





## 1. Introduction

According to the IPCC [1], the provision of biomass for bioenergy and other bio-based products represents an important share of the total mitigation potential for global warming associated with agriculture, forestry, and other land uses. However, the use of bioenergy can lead to either reduced or increased emissions, depending on the scale of deployment, conversion technology, the amount of fossil fuel displaced, and on how and where the biomass is produced [2]. This is particularly true regarding forest biomass. Literature reviews and associated meta-analyses show that whole-tree harvesting (WTH) or forest residue harvesting can reduce soil carbon (C) and nitrogen (N) reserves and other mineral nutrients compared to stem-only harvesting (SOH) [3–5]. Since forest soils are among the largest terrestrial C reservoirs [6,7], C (and N) losses therein caused by certain biomass harvesting practices may become an unforeseen source of atmospheric $CO_2$, thereby impeding the ultimate objective of residual biomass harvesting for energy production, which is to reduce global $CO_2$ emissions.

A striking example of the soil C and N losses associated with biomass harvesting has been observed for WTH compared to SOH in boreal forest stands located in distinct soil landscapes in Quebec, Canada [8]. In this study, we sampled 196 forest plots in the boreal forest in four so-called 'soil provinces' with contrasting soil types. The plots had been harvested 30 years before with either WTH or SOH. Mineral soil C and N stocks (0–60 cm soil depth) were on average 34% lower with WTH than with SOH in the Laurentian and the Mistassini Highlands after 30 years, while there was virtually no difference between the two harvesting intensities in the Appalachians or the Abitibi and James Bay Lowlands. In

the two former soil provinces, the estimated extra biomass harvested with WTH compared to SOH amounted to 21 Mg ha$^{-1}$ on average, which is approximately 10.5 Mg C ha$^{-1}$ and $0.15 \pm 0.01$ Mg N ha$^{-1}$, but this increase in biomass retrieval increased soil C loss to approximately 40 Mg C ha$^{-1}$ and 2 Mg N ha$^{-1}$ [8].

Soil C and N can take several decades to recover following harvest; according to James et al. [9], some Podzol C can only be recovered after at least 75 years. Therefore, research must define the characteristics that distinguish soils "sensitive" to extra forest biomass harvesting from those able to preserve their C and N stocks. Carbon management is the primary impetus for justifying the use of forest biomass as energy feedstock and soil carbon losses should be addressed in forest biomass harvesting policies and guidelines. Most existing guidelines have been established to encourage, or try to ensure, that biomass removal is environmentally sustainable [10]. Most of these guidelines address many issues, such as biodiversity, social values, water and aquatic ecosystems, and soil chemical fertility, acidity, erosion, structure, moisture retention, and compaction. However, the indicators they mention for the long-term storage of C and N derived from soil organic matter often remain vague and general due to the indicators' unknown reactivity or stability [11].

Soil C and N accumulation are influenced by soil texture and other ecological factors, such as vegetation type [7], soil drainage class [12], pedogenesis per se [13,14], and time since major disturbance [15,16]. In addition, soil organic matter (SOM) encompasses a vast array of highly diverse organic residues depending on their size and composition, whether they are in a free state, associated with mineral particle surfaces to varying degrees, or buried within aggregations of mineral particles [17]. These different forms of SOM lead to different properties, which especially influence their stability and lability [18]. In order to improve our understanding of SOM persistence, Lavallee et al. [19], and many others, have proposed separating SOM into two operational physical fractions: "particulate organic matter" (POM) and "mineral-associated organic matter" (MAOM). The two SOM fractions have very distinct compositions and properties [19,20]. For example, mean residence time for POM is from <10 years to several decades, while for MAOM, it ranges from decades to centuries. Therefore, POM is much more labile than MAOM, and is logically more at risk of loss after disturbances such as forest biomass harvesting. In addition, there is a strong relationship between the maximum SOM fraction as MAOM can accumulate in the soil and mineral soil particle size; MAOM is associated with fine particles <20 μm [21,22]. As total SOM is the sum of MAOM and POM, the estimation of potential maximum MAOM allowed us to estimate the corresponding minimum POM fractions from soil particle analyses.

The objective of this study was to find soil properties that could explain the soil C and N differences between WTH and SOH we had found in our earlier study [8], and therefore, to define indicators to identify soils sensitive to C and N loss caused by forest biomass harvesting. In particular, two hypotheses were designed to explain the differences in soil C and N between WTH and SOH:

1. The difference could be related to the existing C and N stocks in the soils;
2. The difference could be related to fine particle content.

In addition, the decomposition process of organic matter is hampered in poorly drained soils [23,24]. Therefore, we also tested if differences in total soil C and N between WTH and SOH were associated with the soil drainage class.

## 2. Materials and Methods

The experimental set-up and relevant soil sampling protocol are briefly described below. More details on forest stand composition and sampling can be found in [8].

### 2.1. Study Area

The study area was mainly located in the balsam fir–white birch bioclimatic domain in eastern Quebec and the southern part of the black spruce–feather moss bioclimatic domain in western Quebec, Canada, between 47° and 50° N latitudes. The study plots included four of the five soil provinces recognized in southern Quebec [25]: the Appalachians (B);

the Laurentians (C); the Abitibi and James Bay Lowlands (D); and the Mistassini Highlands (E) (Supplementary Materials, Figure S1). These soil provinces are distinguished according to the main parent material (geology and geomorphology), altitude (e.g., areas invaded by post-glacial seas), topography (slopes and landforms), soil texture and climate (temperature, rainfall, degree days, etc.; Supplementary Materials, Table S1). Black spruce (*Picea mariana* (Mill.) B.S.P.), balsam fir (*Abies balsamea* (L.) Mill.), and Jack pine (*Pinus banksiana* Lamb.) are the main tree species growing in this boreal forest.

## 2.2. Experimental Design

To determine the effect of harvesting intensity on natural regeneration, from 1982 to 1988, 562 circular plots (0.04 ha each, 11.28 m radius) in a completely randomized design were established just before clear-cut harvesting took place throughout the boreal forest of Quebec. The felling system was either WTH (feller buncher, mechanical harvester and cable skidder, or manual felling and cable skidder) or SOH (manual or mechanical felling and delimbing on-site, then stem extraction by cable skidder). No differences in tree species composition or abundance after natural regeneration were observed between WTH and SOH plots, even up to 10 years after harvesting [26].

In 2011, we selected 196 of these plots among the four soil provinces (n-$_{SOH}$ = 110; n-$_{WTH}$ = 86) based on their accessibility and their similarities regarding pre-harvest stand composition, surface deposit types and depth, and drainage class. Within each soil province, we considered that the WTH and SOH plots had a same pedogenesis and vegetation type, and that there was no difference between pre-harvest and post-regeneration stand composition. Soil characteristics were, therefore, likely to be the main factor explaining the difference in C and N contents between harvesting treatments. The selected plots were revisited for tree measurements and soil sampling from 2011 to 2013, a period representing a median of 30 years after treatment (interquartile range: 2 years).

## 2.3. Field Sampling

The mineral soil was sampled quantitatively at depth intervals of approximately 15 cm, down to 100 cm when possible, with a bipartite 8 cm volumetric root auger at each of the four cardinal points (subsamples every 100 gon, or approximately 16 m). The precise sampling depth was recorded for each sample. When it was not possible to quantitatively sample a soil layer, we used a standard Edelman auger. We also attributed a soil drainage class according to provincial standards [27]. In all, 16 plots were classified as having excessive drainage (Class 1), 96 were well drained (Class 2), 57 were moderately well drained (Class 3), and 27 were imperfectly to poorly drained (Classes 4 and 5).

## 2.4. Laboratory Analyses

We collected a total of 3482 soil samples. All the samples were kept frozen ($-18\,°C$) until preparation for laboratory analysis. The samples were air-dried for at least four days, after which the quantitative samples were weighed individually. The air-dried mineral soil subsamples were passed through a 2 mm mesh sieve. The fine fraction was analyzed for remaining humidity and the concentration of organic matter was assessed with the loss on ignition method. Then, the subsamples of the fine fraction were further ground to 250 µm to determine total C and N with the dry combustion method (LECO CR-412, LECO Corporation, St. Joseph, MI, USA). Since soils did not contain any carbonates, total C was considered as organic C. Particle size analysis was carried out on all 0–15 cm depth soil samples (hydrometer method [28]) and size classes from the Canadian Soil Classification Working Group [29] were used: sand (50–2000 µm); coarse silt (20–50 µm); fine silt (5–20 µm); clay (0–5 µm). The >2 mm fraction of the quantitative mineral soil samples was weighed to determine the proportion of coarser fragments (f) when present.

### 2.5. Computations

Total C and N contents in a given mineral soil layer were calculated following Equation (1)

$$Q_m = 0.1 \times [x] \times D_b \times E_e, \tag{1}$$

where

$Q_m$ is the element content in the mineral soil (kg·ha$^{-1}$);
[x] is the element concentration (mg·kg$^{-1}$) in oven-dried samples;
$D_b$ is bulk density (g·cm$^{-3}$); and
$E_e$ is the effective thickness of the layer (cm), i.e., the thickness of the soil layer without fragments (f).
Effective horizon thickness was calculated following Equation (2)

$$E_e = E \times (1 - f) \tag{2}$$

where

$E_e$ is the effective thickness of the layer (cm);
E is the measured thickness of the layer (cm); f is the coarse fraction of the volumetric sample (>2 mm) (% / 100).

We used a quantitative relationship obtained from the measured Db and SOM concentrations of the quantitative mineral soil samples (n = 1368) to estimate the bulk density of the soil layers that had not been sampled quantitatively (n = 2114) for each great soil texture class (sand, loam, and clay), following Federer et al.'s methodology [30] (Supplementary Materials, Table S2 and Figure S2). We used the *aqp* R package v. 1.17 [31] to compile soil C and N concentrations and stocks, and C/N ratios at fixed depths of 0–15, 15–30, 30–45, and 45–60 cm (measured thickness). C and N mean concentrations and ratios were calculated for the whole depth (0–60 cm), and soil C and N pools in each layer were summed. All soil concentrations and pools are reported for oven-dried mass.

Since no contrasting changes or discontinuities in soil texture were observed down to 60 cm depth at any of our soil sampling points, we considered the soil texture data from the upper sampled layer (0–15 cm) to be representative of the whole 0–60 cm soil sample. The clay fraction of the top mineral layer of forest soils, including the upper B and C horizons, is quite similar throughout Quebec (r = 0.86, *p* < 0.001) [32].

We used the relationship obtained with the boundary-line approach by Feng et al. [22], that is, 0.78 times the percentage of fine soil particles (0–20 μm) to estimate the potential maximum MAOM–C concentrations in our soil samples. The factor 0.78 represents the upper envelope of the relationship between MAOM-C and the percentage of fine soil particles in various soils, so the calculated MAOM–C and –N represent the potential maximal C and N fraction that can be associated with fine particles. To estimate the potential MAOM-N concentrations, we simply divided the MAOM–C concentrations by the average C/N ratio in the 0–60 cm soil layer. Soil POM–C and POM–N were then estimated as follows:

$$[POM-x] = [x] - \min([x], [MAOM-x]) \tag{3}$$

where [x] is observed soil C or N concentrations.

C and N fractions as percentage of total C and N concentrations were also computed. It should be noted that this approach calculates MAOM–C and –N as the potential maximal C and N fractions that can be associated with fine particles, while POM–C and –N represent the minimum C and N fractions that are associated with particulate organic matter.

We calculated the relative loss in C and N soil stocks with WTH compared to SOH as the ratio of the stocks in each individual WHT plot over the average C and N stock values in the SOH plots within a given soil province, as follows:

$$\text{Relative x stock} = 100 \times \text{x stock in WTH / Average x stock in SOH} \tag{4}$$

where x = soil C or N.

*2.6. Statistical Analysis*

We used a generalized linear mixed model to analyze soil C and N fractions and pools for the whole 0–60 cm mineral soil depth range. Harvesting treatments (SOH and WTH), soil provinces, proportion of soil fine particles, and their third-order interactions were considered fixed effects. A second set of analyses was performed using soil C and N pools with harvesting treatment and drainage class, and their interactions, as fixed effects. As some combinations of harvesting treatment and drainage class were missing for this second analysis, we combined them and selected a priori contrasts to test treatment combination effects. The homoscedasticity and normality of sample distributions were verified through residual plot analysis. We then plotted the standardized residuals of the models against all the independent variables to detect possible heterogeneity in their variances. If present, variance heterogeneity was corrected by allowing for different variances per strata. We selected the models for which the variance function structures had the lowest Akaike information criterion (AIC) scores. Adjusted (predicted) means were computed with the *emmeans* R package v. 1.4.2 [33], and statistical comparisons were made through specific a priori contrasts between harvesting treatments within soil provinces. The analyses were performed with the *nlme* R package v. 3.1-141 [34].

We estimated the proportion of fine particles needed and its 95% confidence interval (CI) so that relative soil C and N stocks with WTH attained 90% of the average stock for SOH within a given soil province. We used the *mod_alcc* function of the *soiltestcorr* R package [35] to assess the relationship between relative stocks and soil fine particle content. This function builds an arcsine–log calibration curve based on a standardized major axis; it is a bivariate regression model that assumes that both axes are random variables. To compare the C and N proportions for the sites in the two harvesting treatment groups, we applied chi-square tests for equality of proportions with Yate's continuity correction (*prop.test* function of the *stats* R package) [36].

## 3. Results and Discussion

### 3.1. Soil C and N Stocks with WTH Compared to SOH

Soil stocks ranged from 18 to 268 Mg ha$^{-1}$ for C, and from 0.4 to 13.4 Mg ha$^1$ for N for all the studied plots. For both C and N, some provinces only showed differences between SOH and WTH and when observed, losses were greater for WTH. Figure 1 shows the mean differences in C and N soil stocks between SOH and WTH, and average C and N stocks in SOH by soil province. Average differences in soil C stocks between SOH and WTH ranged from around zero for Soil Provinces B and D ($p \geq 0.407$) to as much as $42 \pm 11$ Mg ha$^{-1}$ in favor of SOH in Soil Provinces C and E ($p \leq 0.004$, Table S3). For Soil Provinces C and E, the degree of soil C loss due to WTH was similar, but the soil C stocks in SOH were quite different (Soil Province C: $148 \pm 10$ Mg C ha$^{-1}$ vs. E: $100 \pm 15$ C Mg ha$^{-1}$, $p \leq 0.007$). A similar pattern was observed for soil N stocks; there was an average difference of $1.92 \pm 0.46$ Mg ha$^{-1}$ in favor of SOH in Soil Provinces C and E ($p \leq 0.006$), although soil N stocks in SOH were distinct (Soil Province C: $6.23 \pm 0.42$ Mg ha$^{-1}$ vs. E: $4.81 \pm 0.57$ Mg ha$^{-1}$; $p = 0.029$). Meanwhile, in Soil Provinces B and D, no difference in soil N stocks was found between SOH and WTH ($p \geq 0.582$). These results question the hypothesis that the soil C and N loss attributable to WTH is directly proportional to the initial soil C and N stocks.

A limitation of this study was the lack of pre-harvest soil data. However, the high replication of sites compensated for this issue, at least partially. In general, SOH leads to an increase in SOM pools as compared to non-harvested sites, while WTH does the opposite [37]. Therefore, the differences in C and N stocks between WTH and SOH sites that we observed in soils of the two Soil Provinces C and E can be attributed, at least in part, to an increase in these pools in SOH sites and a loss in WTH sites.

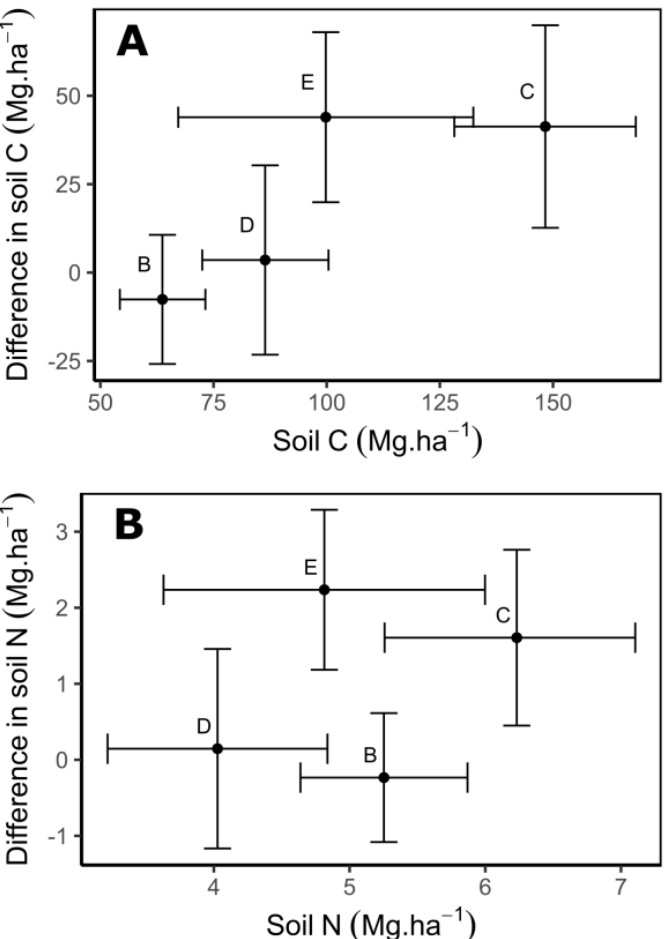

**Figure 1.** Differences in soil (**A**) C and (**B**) N stocks (0–60 cm soil depth) between stem-only (SOH) and whole-tree (WTH) harvesting as a function of soil C and N stocks for SOH by soil province. Adjusted means ± 95% confidence intervals. Soil provinces: B: Appalachians; C: Laurentians; D: Abitibi and James Bay Lowlands; E: Mistassini Highlands.

### 3.2. Relationship between Soil C and N Stocks with WTH and Fine Particle Content

The four soil provinces were distinct in terms of soil particle content in the fine earth (Figure 2). Soil Provinces C and E were the richest in sand (mean ± SD: 69 ± 12% and 60 ± 11%, respectively) and the poorest in clay (9 ± 3% and 8 ± 5%, respectively). Conversely, Soil Provinces B and D were the poorest in sand (38 ± 11% and 43 ± 28%, respectively) and the richest in clay (25 ± 6% and 33 ± 26%, respectively). In terms of fine particles <20 μm, Soil Provinces B and D were the richest (49 ± 10% and 48 ± 29%, respectively) and Soil Provinces C and E the poorest (18 ± 6% and 25 ± 9%, respectively) ($p < 0.001$).

For Soil Provinces B and D, our analysis of the relationship of soil C and N stocks in the 0–60 cm mineral soil depth as a function of soil province, harvesting treatment, and soil fine particle content revealed no significant effect of harvesting treatment or of soil fine particle content, nor did their interaction have any effect ($p \geq 0.432$). However, a different pattern was observed for Soil Provinces C and E, where a significant interaction between fine particle content and harvesting treatment was apparent for soil C, and to a lesser extent, for soil N ($0.006 \leq p \leq 0.093$). The differences in total soil C and N stocks between WTH and SOH, expressed as relative stocks for WTH over average stocks for SOH, reached 90% of the average stocks for SOH within each soil province when the soil fine particle content reached $\geq 30\%$ (Figure 3). Its confidence intervals (CI95%) range from 26.1% to 34.4% for 90% soil C stocks and from 28.0% to 37.4% for 90% soil N stocks.

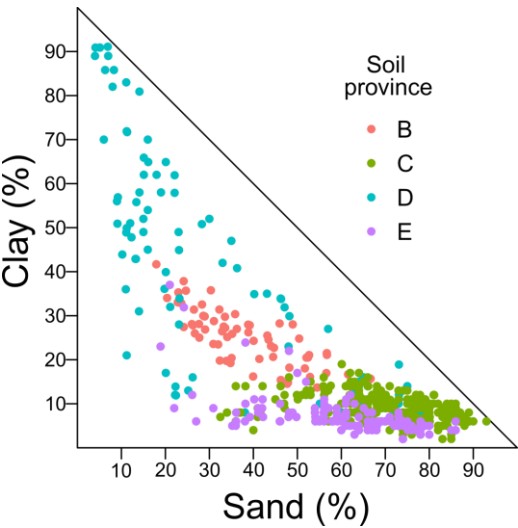

**Figure 2.** Soil texture diagram (Canadian soil texture triangle) with forest sampling points according to soil province. Soil provinces: B: Appalachians; C: Laurentians; D: Abitibi and James Bay Lowlands; E: Mistassini Highlands.

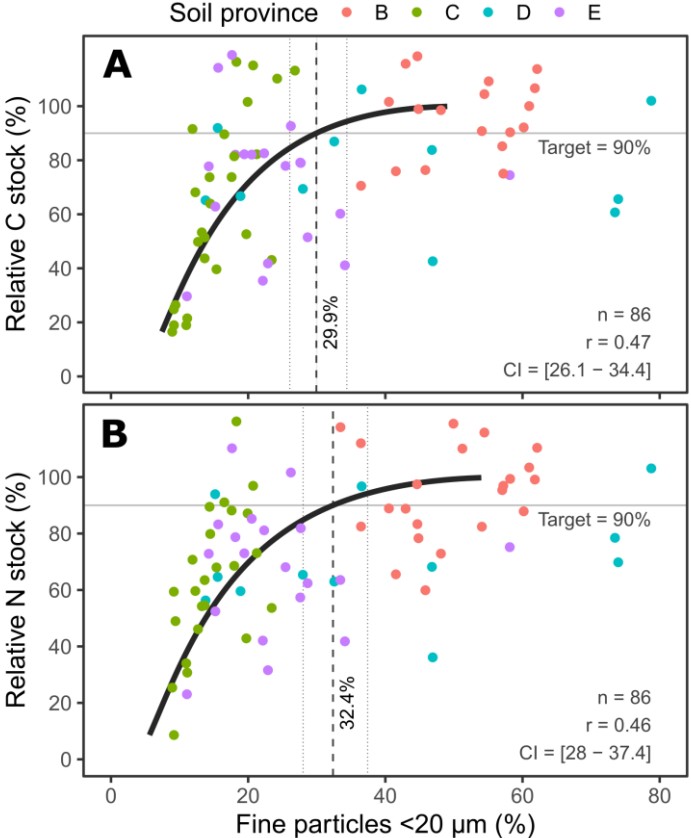

**Figure 3.** Relationship between soil fine particle content and (**A**) relative soil C stocks and (**B**) relative soil N stocks (0–60 cm soil depth) after whole-tree harvesting (WTH) (relative to average stem-only harvesting, SOH) by soil province. The vertical dashed line represents estimated fine particle content (and its 95% confidence intervals (CI)) required to reach 90% of the relative soil C and N stocks after SOH with the arcsine–log calibration curve; r is the square root of the proportion of variation explained by the curve. Soil provinces: B: Appalachians; C: Laurentians; D: Abitibi and James Bay Lowlands; E: Mistassini Highlands. Curves: relative soil C stocks = $100(\sin(-0.776 + 0.596 \ln(x))^2)$; relative soil N stocks = $100(\sin(-0.637 + 0.543 \ln(x))^2)$.

Our results are in agreement with the meta-analysis by Wan et al. [38], who reported a general increase in mineral soil C losses with WTH as compared with SOH only for forest soils with a clay content of < 20%. This result is very similar to our threshold of 30% for fine particles <20 μm, which corresponds to a clay content of 12% to 17%. In similar boreal forests, Morris et al. [39] found that 20 years after various harvesting intensities, mineral soil C and N stock losses were greater only in coarser-textured sandy soils.

Although deadwood in SOH sites does not appear to be a significant source of SOM in coarse-textured soils [40], the slash left on the ground after SOH can be abundant, averaging 34 Mg ha$^{-1}$, compared to 5 Mg ha$^{-1}$ for WTH, for woody debris ≤7 cm in diameter [41]. Abundant slash may play a significant role in preserving POM in soils partly through its insulating and cooling properties [42–46]. The forest soil temperature regime and C balance can change more drastically after WTH compared to SOH [47–49], with consequences lasting several years. This is also true when the soil surface is exposed [50,51]. In addition, the natural forest regeneration established after harvesting may have played a role in SOM accumulation. Although the natural tree regeneration growth and abundance were not affected for up to 10 years after the harvesting treatments [26], lower tree growth, in the order of 15% in stem basal area increment, was observed for WTH compared to SOH in Soil Provinces C and E during the first 30 years [8]. This lower tree growth and the resulting reduced below and aboveground SOM input may have contributed to the differences in soil C and N stocks between WTH and SOH after 30 years.

*3.3. The Role of C and N Fractions*

The effect of soil fine particles on soil C and N stocks can be explained by the status of SOM, which can either be rather stable in the MAOM fraction, or more labile in the POM fraction [19,20]. The role of MAOM in soil C and N preservation is shown in Figure 4. In this figure, we plotted potential MAOM-C and MAOM-N concentrations estimated by the proportion of fine particles <20 μm in the soil against actual soil C and N concentrations according to soil provinces. The lines represent a 1:1 relationship; data above that line indicate that the maximal capacity to sequester C and N in the MAOM fraction is greater than the actual C and N measured. The figure indicates that almost all the sites in the two soil provinces with no loss of soil C and N attributable to WTH (i.e., Soil Provinces B and D) are above the 1:1 line.

This means that most of the soil C and N could potentially be locked in the MAOM fraction. In fact, Soil Provinces B and D have the capacity to sequester two to four times more C and N in their fine particles than their actual current concentrations.

That is not the case for Soil Provinces C and E, where losses of soil C and N attributable to WTH compared to SOH were observed. In these soils, a large part of the C and N could not be sequestered by the MAOM fraction, and therefore, remaining as soil POM, which, theoretically, is more prone to decomposition. In Soil Province C (n = 69), 51% of the sites did not have the potential to sequester all the C and N in their fine particle fraction (i.e., points falling below the 1:1 line). In Soil Province E, 27% of the sites did not have sufficient sequestration potential.

Obviously, it may be unrealistic to believe that the potential maximal MAOM fraction was attained in these soils because the 0.78 factor converting fine soil particle content to maximum MAOM-C fraction may not apply to all of them. However, applying the exact factor would not change the general pattern observed among soil provinces; that is, Soil Provinces B and D had a higher capacity to sequester SOM in a stable form than Soil Provinces C and E because of their higher soil fine particle content.

Since there is no theoretical limit to the accumulation of (more labile) POM in soils [52,53], the potential for soil C and N loss should be influenced more directly by the amount of POM rather than by total SOM. We examine this hypothesis in Figure 5, which shows the relationships between the estimated minimum POM–C and –N fractions and their corresponding actual C and N stocks according to soil provinces and harvesting intensity. Minimum POM–C and –N fractions increased with their corresponding total stocks only

in Soil Provinces C and E, where there were C and N losses with WTH compared to SOH. Meanwhile, there was no increase in minimum POM–C and –N fractions with their corresponding total stocks in Soil Provinces B and D, where no such losses in C and N occurred when WTH was compared to SOH.

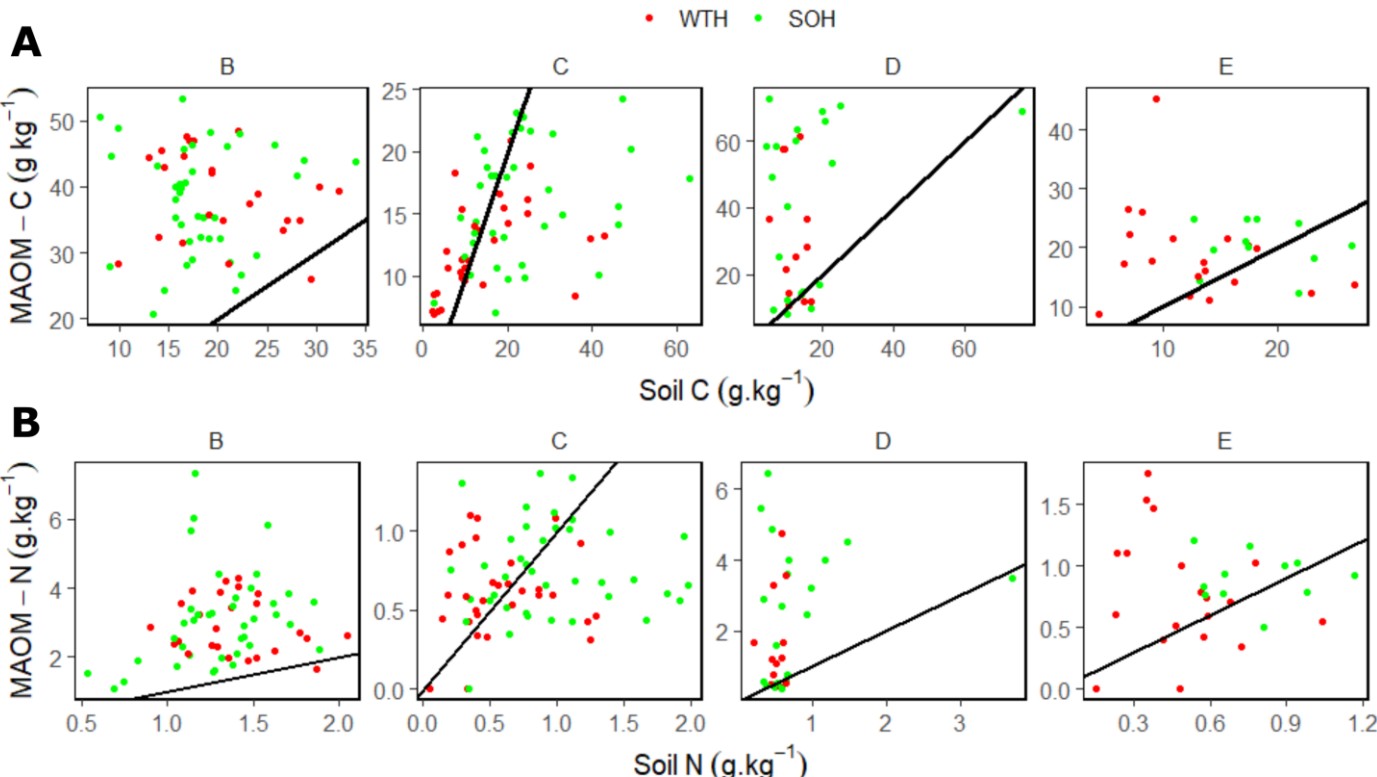

**Figure 4.** Potential mineral-associated organic matter (MAOM) (**A**) C concentrations and (**B**) N concentrations as a function of measured C and N concentrations in the soils, according to forest harvesting treatment and soil province (n = 196). Lines represent a 1:1 relationship. Harvesting treatments: whole-tree harvesting (WTH); stem-only harvesting (SOH). Soil provinces: B: Appalachians; C: Laurentians; D: Abitibi and James Bay Lowlands; E: Mistassini Highlands.

Therefore, it appears that actual soil C and N losses attributable to WTH are not entirely associated with their initial total stocks in the soil, but rather that they are more related to the abundance of soil POM, at least in these boreal forest soils. This conclusion is in agreement with what Achat et al. [3] report in their meta-analysis; they state that soil C losses in the upper mineral layers after WTH increased with increasing initial soil C stocks; this is probably because a higher fraction of the C stocks is held as POM in soils with higher soil C stocks, as we found for Soil Provinces C and E.

*3.4. Relationship between Soil C and N Stocks with WTH and Drainage Conditions*

Soil C and N stocks differed significantly between SOH and WTH for imperfectly or poorly drained soils: −50% for both C and N stocks (drainage class 4–5, $p = 0.043$, Figure 6). Soil N stocks also differed between SOH and WTH in moderately well-drained soils (−30%, drainage class 3, $p = 0.001$). We first assumed that the poorer the drainage, the higher the soil fine particle content. However, our initial analyses revealed the same trends even when the data were split by soil fine particle content (<30% and ≥30% of particles <20 µm). In addition, we found no relationship between drainage class and soil fine particle <20 µm content within the soil provinces ($Chi^2 \leq 2.92$, $p \geq 0.232$), excluding a possible confounding factor. Nevertheless, our results are based on relatively few observations (n = 10 for the WTH/drainage class 4–5 combination, n = 17 for the SOH/drainage class 4–5 combination), so these findings must be taken with caution.

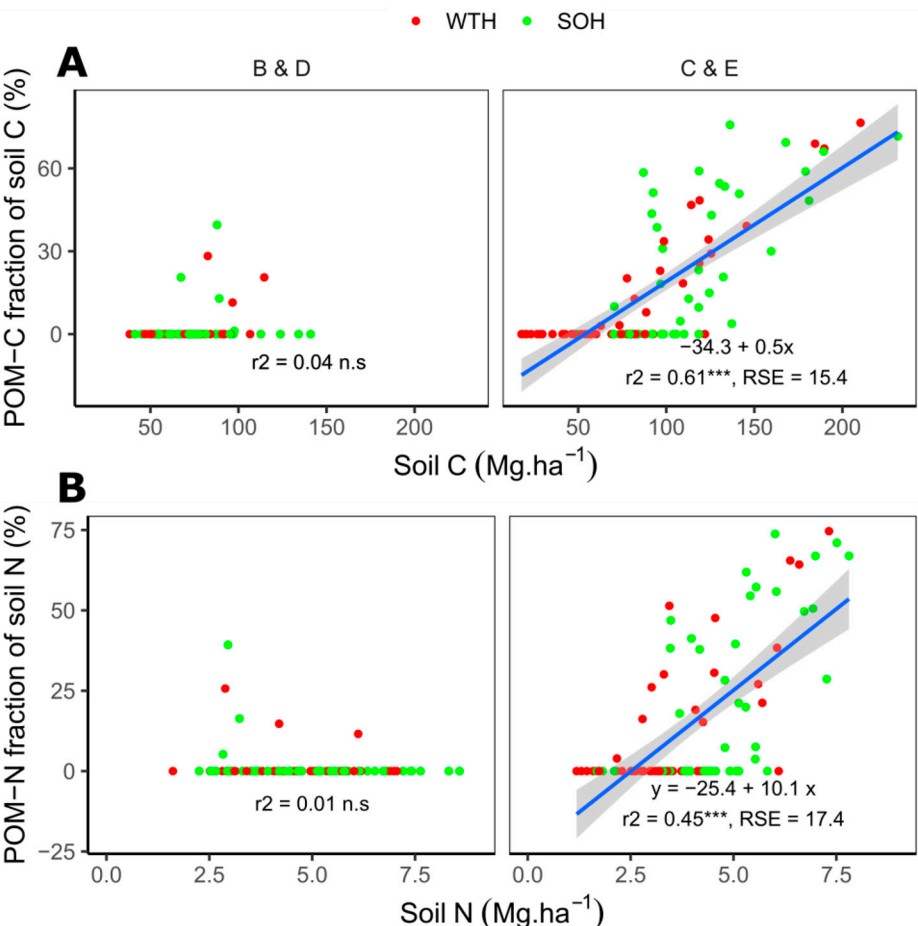

**Figure 5.** Estimated minimum soil (**A**) C fractions and (**B**) N fractions in the form of particulate organic matter (POM) as a function of measured soil C and N stocks for whole-tree harvesting (WTH) and stem-only harvesting (SOH) in Soil Provinces B and D (left panels) and C and E (right panels). The shaded area around the regression lines represents ± 1 SE. Soil Provinces B and D: Appalachians, and Abitibi and James Bay Lowlands; C and E: Laurentians and Mistassini Highlands.

The lower C and N stocks we observed in the WTH plots associated with poor soil drainage conditions suggest that SOM might have been higher in the POM fraction in those soils due to reduced SOM decomposition [54]. Indeed, slower-draining soils usually accumulate more C than well-drained soils for similar soil textures [12,55]. In well-drained soils, fine particles are a major factor of SOM stabilization. In poorly drained soils, on the other hand, cold temperatures, the lack of oxygen and substrate quality interact to stabilize SOM [23,24]. Therefore, as this SOM is not protected by mineral particles and other ligands, poorly drained soils may be prone to C and N losses in case of a major soil surface disturbance, such as WTH. The persistence of SOM in soils is ultimately controlled by interacting microscale processes shaped by the moisture regime [56]. More research is needed in order to better understand the impact of additional biomass harvesting on soil C and N stability in poorly drained soils.

*3.5. Management and Policy Implications*

Our analyses indicate that the differences in total soil C and N between SOH and WTH were not directly proportional to their initial stocks, but were more related to soil fine particle content and associated SOM. These two factors made a clear differentiation between the soil provinces having preserved, or lost, their soil C and N after WTH. Therefore, soil C and N sensitivity to loss after additional biomass harvesting appears to be closely related to the amount of fine mineral particles the soil contains, as stated in our second hypothesis.

Finally, differences in soil C and N stocks between SOH and WTH also appear to be associated with a particular soil drainage class, namely imperfectly or poorly drained soils. However, more observations are needed on the impact of WTH on soils in poor drainage conditions before making more general recommendations concerning additional biomass harvesting.

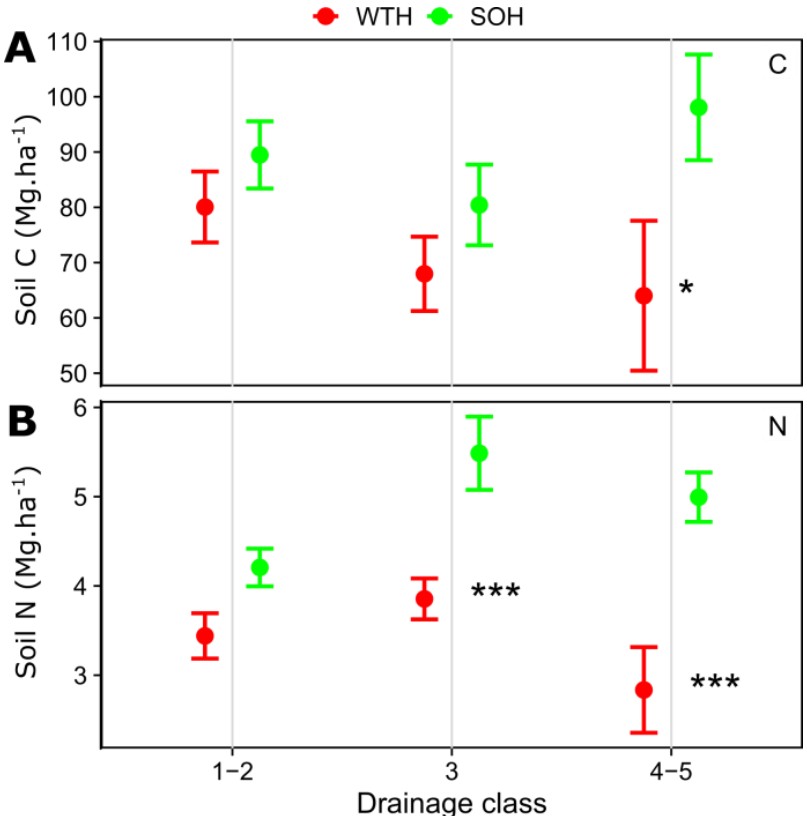

**Figure 6.** Relationships between (**A**) soil C and (**B**) soil N stocks (0–60 cm) and drainage class according to forest harvesting treatment (WTH: whole-tree harvesting; SOH: stem-only harvesting) after 30 years. * indicates a significant difference between harvesting treatments at probability * $p \leq 0.05$ and *** $p \leq 0.001$.

Currently the forest sustainability policy for upland forests in Quebec relies only on soil acidification criteria to classify soil sensitivity to additional biomass harvesting. Interestingly, the soil provinces where critical loads of acidity are exceeded [57] are the same soil provinces we found to be the most sensitive to C and N losses after WTH compared to SOH. Existing policy targets forest ecological types with certain basic site attributes (e.g., coarse-textured, thin or organic soils). No specific guidelines for soil C and N preservation exist. The analyses presented in this paper allow us to propose some guidelines that may help fill this void and to preserve soil C and N stocks from the impacts of additional biomass harvesting in boreal forests in the long term. We found that a minimum threshold of 30% fine particles <20 μm ensures that WTH will have a similar impact on soil C and N to SOH in the long term. Thirty percent of fine particles <20 μm corresponds to a threshold of 45% silt + clay (Supplementary Materials, Figure S3). Limiting additional biomass harvesting to sites where forest soils contain ≥45% silt + clay and have a drainage class <4 (i.e., upland soils) would help the forest sector preserve soil C and N from potential losses. These guidelines could be applied readily as soil drainage and texture maps are now available for the whole forest territory in Quebec [32,58,59]. It is, therefore, possible to determine soil sensitivity to C and N losses after additional biomass harvesting at the scale of the forest stand in Quebec's boreal forest.

The extrapolation of these results to temperate broad-leaved forests is not straightforward. In general, coniferous forests store more SOM than do broad-leaved forests, and have a higher proportion of soil C in the POM fraction [7]. In coniferous forests, the MAOM-C fraction, on the other hand, is comparable to that of broad-leaved forests, so in broad-leaved forests, soil C and N should be less impacted by forest biomass harvesting [60]. This hypothesis remains largely undocumented, however.

## 4. Conclusions

In this study, we hoped to find soil properties that could explain the higher losses of soil C and N at sites subjected to WTH compared to sites subjected to SOH 30 years before. We found that lower soil C and N stocks with WTH than with SOH were not directly associated with corresponding initial stocks, but to the abundance of soil fine particles. C and N stocks were maintained after WTH in soils with at least 30% of fine particles <20 μm. A low fine particle <20 μm content leads to the saturation of the MAOM fraction and a greater abundance of POM in the soil, and this could explain why soil C and N were lower in WTH soils compared to SOH soils in the Laurentians and Mistassini Highlands soil provinces. Conversely, a higher fine particle <20 μm content increases the ability of the MAOM fraction to store C and N; this explains why there was no difference in soil C and N between WTH and SOH in the Appalachians and Abitibi and James Bay Lowlands provinces. Furthermore, imperfectly and poorly drained soils appeared to be sensitive to additional biomass harvesting regardless of soil texture, but this finding warrants further investigation of the mechanisms at play. In the context of the fight against climate change, it is important to stress that soil C and N reserves should be conserved, and if possible even enhanced, by forest operations in order to reduce greenhouse gases in the atmosphere.

**Supplementary Materials:** The following supporting information can be downloaded at: https://www.mdpi.com/article/10.3390/soilsystems7020039/s1, Table S1. Characteristics of the four soil provinces in Quebec included in the study. Table S2. Parameter estimates for the relationship between soil bulk density (Db) and organic matter (OM) concentrations according to broad soil texture groups. Table S3. Total carbon (C) and total nitrogen (N) stocks in the first 60 cm of soil (forest floor excluded) as a function of harvesting treatment (stem-only harvesting (SOH) and whole-tree harvesting (WTH)) in four soil provinces in Quebec: B = Appalachians; C = Laurentians; D = Abitibi Lowlands; E = Mistassini Highlands. Data presented are model-adjusted means ± SE. Figure S1. Map of sampling plot numbers and locations. Colored areas show the different soil provinces in Quebec (Canada). Green lines delimit vegetation zones according to bioclimatic domains: SM-NH: sugar maple northern hardwoods; BF-YB: balsam fir–yellow birch domain; BF-WB: balsam fir–white birch domain; BS-FM: black spruce–feather moss domain. Figure S2. Relationship between soil bulk density (Db) and organic matter (OM) concentrations according to broad soil texture groups. Lines show values predicted by the model with a 95%CI. Figure S3. Relationship between soil fine particle content (silt + clay) and the relative average C and N stocks after whole-tree harvesting (WTH) (relative to stem-only harvesting, SOH) by soil province. The dashed line in the arcsine–log calibration curve represents the silt + clay content (95% confidence intervals (CI)) required to reach 90% of the soil C and N stocks after SOH [35]. Curves: relative C stocks = $100(\sin(-1.384 + 0.698 \ln(x))^2)$; relative N stocks = $100(\sin(-1.191 + 0.636 \ln(x))^2)$.

**Author Contributions:** Conceptualization, R.O., N.K. and I.B.; methodology, R.O.; validation, N.K. and I.B.; formal analysis, R.O.; writing—original draft preparation, R.O.; writing—review and editing, N.K. and I.B.; funding acquisition, R.O., N.K. and I.B. All authors have read and agreed to the published version of the manuscript.

**Funding:** This research was funded by the Ministère des Ressources naturelles et des Forêts du Québec (project No. 142332122), by the Quebec 2030 Plan for a Green Economy, and by Le STUDIUM, Loire Valley Institute for Advanced Studies, Orléans & Tours, France, for a research fellowship to R.O. at INRAE.

**Data Availability Statement:** The soil samples for further analyses and data presented in this study are available on request from the corresponding author; data are also available at https://doi.org/10.5281/zenodo.7654866.

**Acknowledgments:** The authors gratefully acknowledge the research fellowship support by Le STUDIUM, Loire Valley Institute for Advanced Studies. We also thank Vicki Moore for editing the English in the manuscript.

**Conflicts of Interest:** The authors declare no conflict of interest. The funders had no role in the design of the study; in the collection, analyses, or interpretation of data; and in the writing of the manuscript.

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
