# Peer review of "Soil Texture Explains Soil Sensitivity to C and N Losses from Whole-Tree Harvesting in the Boreal Forest"

_soilsystems, doi:10.3390/soilsystems7020039_

Round 1

Reviewer 1 Report

The study deals with soil C and N stocks and soil particle size classes 30 years past whole tree harvest (WTH) and stem only harvest (SOH) in Quebec Canada. Such long-term studies are of high value. The outcome is interesting, suggesting much higher C losses from WTH in soil with lower fine particle (clay, silt) content. I have some suggestions to further develop the paper:

1.    The paper starts with the hard measured facts, which are soil C and N stocks and soil texture. Then the concept of maximal potential carbon sorption to soil minerals is introduced and calculation of potential maximal mineral associated carbon (MAOM-C) and minimal particulate organic carbon (POC) content follows. I am quite skeptical about that. MAOM-C and POC were not measured (why?) but just estimated using a factor (0.78) extracted from the study of Feng et al. Biogeochemistry 2011. I had a look at the Feng paper and found that the story isn’t that simple. The factor varies substantially with the method it is calculated with, but also among different soils in different ecosystems and among minerals with different loadings. Hence picking one value (0.78) out of a range between 0.30 until >1 (Tab 3 in the Feng paper) seems a bit like cherry picking. The factor size has huge influence on the results and their interpretation. I therefore suggest removing the maximum MAOM-C and minimum POC calculations from the manuscript. The importance of MAOM-C can easily be discussed without that – or to apply a density fractionation to a subset of the samples to determine the real MAOM-C and POC contents.

2.      C stocks were significantly lower under WTH than under SOH in two soil provinces and the absolute difference was huge with around 40 tons C per hectare or up to one third of the total C stock. There could be more emphasizes on providing potential explanations for the huge losses in the discussion. One reason could be differences in soil C loss (respiration, leaching) – respiration is mentioned already. Another reason, however, could be differences in C input during the 30 years past harvest. I miss this point in the discussion section. It is stated that tree regeneration was not affected 10 years after harvest, but what happened thereafter? How did the stands look like 30 years after harvest? Was C input similar at the different treatment sites? Was there any assessment done on this?

3.      I wonder how and why the relatively small differences among biomass removal during WTH and SOH could result in such huge differences in soil C stocks over time. Probably leaching or even surface runoff of POC from open soil surfaces might have played a role (was the terrain flat or were the forest located on slopes?). Slash might intercept a portion of the precipitation input and avoid leaching of POM, or it might prevent massive surface runoff and associated colloidal POC losses during strong rain events. Just an idea, but could be looked up in the literature and added to the discussion. I have in mind that clear-cutting can result in severe humus loss from steep mountain slopes by runoff. With this regard it would also be very interesting if the C was lost from the topsoil or organic layer or from the mineral soil in deeper soil layers. Can you provide information on C distribution with soil depth?

4.      Maybe this goes too far, but authors mention that several factors can influence the effect of harvest on soil C stocks (L102: parent material, altitude, climate, topography, texture; L389: acidity). A multifactorial analyses of the CN data using all these parameters would reveal which one really determines the effect. Are all these parameters available for the sites? In the present manuscript co-variation of texture (fine particle density) with any of the other parameters cannot be excluded. A multifactorial approach would shift the study to another level, but at least it could be mentioned in the discussion that other variables as texture could as well be the cause for the different response to harvest procedures among the soil provinces (for acidity at least this is done in L389 already).

5.      A limitation of the study is the lack of pre-harvest soil data. However, the good replication of sites can, at least partially, compensate this issue. Nonetheless the limitation should be mentioned in the discussion section.

Line comments:

L76-79: unclear what “minimum POM” means here because the term is not introduced before

L349: showed revealed

L387: (choose one of the words) ?

Reviewer 2 Report

Authors showed that soil organic C and total N response to soil texture and forest management. This study conducted at a 30 years treatment, and it will be important to get a general conclusion. Huge of soil samples were collected and analyzed, well organized. I recommended a substantial revised is needed.

Specific comments:

1.     I recommended author to rewrite the part of abstract, it should be more logically and reasonable. Line 17-19, I think its improper to explain the terms in abstract, It doesn't make sense. It will be better to demonstrate the results, conclusions or highlight implications.

2.     Line 45-50, these results are part of this study? If not, please insert a literature.

3.     Soil C should more specific Soil organic C, and soil N change to soil total N?

4.     Line 51, “according to [9]” is incorrect presentation.

5.     Line 84-87, I suggest to raise two hypothesis

6.     Line 103, authors should tell us how to define as soil texture, is there references? What’s the different of soil property.

7.      Line 144-147 authors should briefly state the method of particle size analysis, although authors mentioned Soil Classification such as: “sand (50–2000 µm); coarse silt (20–50 µm); fine silt (5–20)……..”. Generally, Soil fractionation was done using a procedure of Cambardella and Elliott (1992), soil should be shaken in sodium hexametaphosphate solution, after that, could be defined as POC OR MAOC. please declare what’s your detail methods.

8.     Internal figures SHOULD be numbered a, b, c, d ect. and indicated the specific caption in figure legends, thus make it readable.

9.     I suggest authors show us the soil carbon (C) distributions in size fractions.

Round 2

Reviewer 1 Report

Authors adressed al points and questions raised. From my opinion the ms is ready for publication.